# Disruption of Atrial Rhythmicity by the Air Pollutant 1,2-Naphthoquinone: Role of Beta-Adrenergic and Sensory Receptors

**DOI:** 10.3390/biom14010057

**Published:** 2023-12-31

**Authors:** Antonio G. Soares, Simone A. Teixeira, Pratish Thakore, Larissa G. Santos, Walter dos R. P. Filho, Vagner R. Antunes, Marcelo N. Muscará, Susan D. Brain, Soraia K. P. Costa

**Affiliations:** 1Departamento de Farmacologia, Instituto de Ciências Biomédicas, Universidade de São Paulo, Av. Prof Lineu Prestes, 1524, São Paulo 05508-000, SP, Brazil; antonius@alumni.usp.br (A.G.S.); mone@usp.br (S.A.T.); larissagonzagas@usp.br (L.G.S.); muscara@usp.br (M.N.M.); 2Department of Integrative Biology and Pharmacology, University of Texas Health Science Center at Houston, 6431 Fannin St., Houston, TX 77030, USA; 3Section of Vascular Biology and Inflammation, School of Cardiovascular Medicine and Sciences, BHF Cardiovascular Centre of Research Excellence, King’s College London, Franklin-Wilkins Building, London SE1 9NH, UK; pthakore@med.unr.edu; 4Fundação Jorge Duprat Figueiredo de Segurança e Medicina do Trabalho, Ministério do Trabalho e Previdência Social, Rua Capote Valente, nº 710, São Paulo 05409-002, SP, Brazil; walterpf@alumni.usp.br; 5Departamento de Fisiologia e Biofísica, Instituto de Ciências Biomédicas, Universidade de São Paulo, Av. Prof Lineu Prestes, 1524, São Paulo 05508-000, SP, Brazil; antunes@icb.usp.br

**Keywords:** particulate matter, air pollution, transient receptor potential, 1,2-naphthoquinone, atrial fibrillation, ECG, adrenergic receptor, heart failure, mouse

## Abstract

The combustion of fossil fuels contributes to air pollution (AP), which was linked to about 8.79 million global deaths in 2018, mainly due to respiratory and cardiovascular-related effects. Among these, particulate air pollution (PM2.5) stands out as a major risk factor for heart health, especially during vulnerable phases. Our prior study showed that premature exposure to 1,2-naphthoquinone (1,2-NQ), a chemical found in diesel exhaust particles (DEP), exacerbated asthma in adulthood. Moreover, increased concentration of 1,2-NQ contributed to airway inflammation triggered by PM2.5, employing neurogenic pathways related to the up-regulation of transient receptor potential vanilloid 1 (TRPV1). However, the potential impact of early-life exposure to 1,2-naphthoquinone (1,2-NQ) on atrial fibrillation (AF) has not yet been investigated. This study aims to investigate how inhaling 1,2-NQ in early life affects the autonomic adrenergic system and the role played by TRPV1 in these heart disturbances. C57Bl/6 neonate male mice were exposed to 1,2-NQ (100 nM) or its vehicle at 6, 8, and 10 days of life. Early exposure to 1,2-NQ impairs adrenergic responses in the right atria without markedly affecting cholinergic responses. ECG analysis revealed altered rhythmicity in young mice, suggesting increased sympathetic nervous system activity. Furthermore, 1,2-NQ affected β1-adrenergic receptor agonist-mediated positive chronotropism, which was prevented by metoprolol, a β1 receptor blocker. Capsazepine, a TRPV1 blocker but not a TRPC5 blocker, reversed 1,2-NQ-induced cardiac changes. In conclusion, neonate mice exposure to AP 1,2-NQ results in an elevated risk of developing cardiac adrenergic dysfunction, potentially leading to atrial arrhythmia at a young age.

## 1. Introduction

Atrial fibrillation (AF), the most prevalent cardiac arrhythmia, poses a significant burden on patients and healthcare systems [1,2]. It stems from hyperexcitation in specific atrial regions, leading to desynchronized atrial contractions and contributing to elevated morbidity and mortality rates. Associated risks include thromboembolism, heart failure, and stroke [3,4,5,6].

While the association between ambient air pollutants (AP) and atrial fibrillation (AF) is still a matter of debate, several systematic reviews and meta-analyses support a link between AP effects and cardiovascular diseases, including AF [7,8,9,10]. This association, alongside cardiopulmonary issues, is estimated to have contributed to 8.79 million deaths worldwide in 2018 [11], especially affecting vulnerable subpopulations.

The 2012 ESCALA study (Estudio de Salud y Contaminación del Aire en Latinoamérica) found that infants and young children are more vulnerable to PM10 and ozone (O3) than the general population in Chile, Brazil, and Mexico [12]. Short-term exposure to pollutants, including PM2.5, sulfur dioxide (SO_2_), and nitrogen dioxide (NO_2_), increases the likelihood of AF, while prolonged exposure to PM2.5, PM10, SO2, NO2, and carbon monoxide (CO) significantly contributes to AF incidence in healthy populations. Studies also show a higher risk of mortality and stroke in pre-existing AF patients exposed to elevated PM levels [10]. Exposure to PM2.5 is linked to new-onset AF, especially in males over 60 years old with obesity and a history of hypertension or myocardial infarction [13]. Recent research links AP and coronavirus disease (COVID-19) outcomes, with exposures to PM2.5 and NO_2_ potentially influencing COVID-19 severity and mortality, particularly in patients with pre-existing cardiovascular diseases, such as AF [14]. PM2.5 poses a significant global risk factor for cardiovascular morbidity and mortality, causing damage to the cardiovascular system [15,16,17,18]. Once PM2.5 infiltrates the alveoli and enters the bloodstream, lungs, and heart [19,20,21,22,23], it induces damage to the cardiovascular system. This damage encompasses ischemic stroke, vascular dysfunction, hypertension, atherosclerosis, myocardial infarction, and cardiac arrhythmias [23,24,25,26,27].

There are many different and complex mechanisms underlying the circulatory disruptions and lung inflammation caused by PM2.5. According to a number of published studies, PM causes airway inflammation by triggering sensory neuronal pathways, particularly through the cation channels TRPV1 and TRPA1 [28,29]. Sensory C neurons, as well as a variety of immune and non-immune cells, express TRP receptors [30,31]. The generation of cytokines was seen in BEAS-2B cells exposed to residual oil fly ash particles (ROFA) as a result of the activation of TRPV1, which is sensitive to both pH and capsaicin [32]. In agreement, PM-induced airway inflammation affects TRPV1 and acid-sensitive irritant receptors on nerve terminals and somatosensory cell bodies [33].

Additionally, PM2.5 influences platelet activation and DNA methylation [34]. Inhaled fine PM2.5 might prompt the release of pro-thrombotic and inflammatory mediators into the blood circulation, leading to endothelial cell injury [35], immune dysfunction [36,37], and up-regulates TRP receptors and pro-inflammatory mediators, such as cytokines and neuropeptides [36,38,39,40,41].

PM2.5 can directly translocate into the circulation by crossing the lung–blood barrier and interacting with the vasculature. This interaction prompts various detrimental effects, including vasoconstriction, endothelial dysfunction, pro-coagulant/thrombotic actions, increased release of cell adhesion molecules and chemokines, as well as an amplified generation of oxidative stress markers in the vascular endothelium [19,42,43,44]. Furthermore, exposure to PM2.5 has been linked to the activation of the renin–angiotensin–aldosterone system (RAAS) [44,45] and disruption of the autonomic nervous system [43,46], which plays a crucial role in regulating heart rate and blood pressure.

Notably, fine PM poses a substantial risk due to its impact on cardiovascular and pulmonary health. Additionally, it affects human health due to chemical contaminants adsorbed on its surface, thereby adding strength to the hazardous nature of PM [47,48]. Unfortunately, PM2.5 contains quinones, such as 1,2-naphthoquinone (1,2-NQ), a polycyclic aromatic compound formed as a metabolite of naphthalene. In our recent review, we have outlined how 1,2-NQ and related chemicals can initiate and exacerbate harmful pathophysiological processes, leading to tissue and cell damage [48].

Though 1,2-NQ is known to be harmful, its potential has been poorly investigated. Our previous studies have shown that fine PM2.5, diesel exhaust particle (DEP)-induced airway inflammation is significantly influenced by increased levels of 1,2-NQ, acting through neurogenic mechanisms involving the up-regulation of TRPV1 and activation of sensory fibers [38]. Our previous studies used a similar approach to this study and exposed rodents to 1,2-NQ at a postnatal stage, which promoted increased susceptibility to pulmonary allergic inflammation in the adult period. We found that this occurs through a mechanism dependent on the upregulation of co-stimulatory molecules, leading to amplified cell presentation, Th2 polarization, and enhanced levels of LTB4, humoral response, and Th1/Th2 cytokines [49]. More recently, we have shown that sub-chronic neonatal mouse exposure to 1,2-NQ increases adipose tissue inflammation, paralleled by disrupting energy homeostasis, partially mediated by TNFR1 and TLR4-mediated mechanisms [50].

Numerous studies have established a correlation between exposure to PM and cardiac disturbances, such as arrhythmias [51,52,53,54]. However, there remains a gap in the evidence connecting the exposure to the electrophilic contaminant 1,2-NQ found in environmental PM with arrhythmic events. Thus, this study was initiated to probe the impact of early acute exposure to 1,2-NQ. Our findings indicate that premature exposure to 1,2-NQ disrupts adrenergic response while leaving cholinergic responses unaffected. ECG analysis shows altered rhythmicity, indicative of heightened activity in the sympathetic nervous system. Moreover, administration of 1,2-NQ impedes the positive inotropic effect induced by β1-adrenergic receptor agonists in isolated atria, a response that can be reversed by metoprolol, a β1 receptor blocker. Notably, the cardiac changes induced by 1,2-NQ are mitigated by capsazepine, a TRPV1 blocker, but not by a TRPC5 antagonist. We reveal that neonatal exposure of mice to 1,2-NQ elevates the risk of developing cardiac adrenergic dysfunction, potentially leading to atrial arrhythmia at a young age. These findings underscore the significance of understanding the impact of environmental contaminants on cardiac health, particularly during early developmental stages.

## 2. Materials and Methods

### 2.1. Animals

All animal protocols and experiments were conducted with the approval of the local Ethical Committee for Animal Research (48/2016-CEUA) in compliance with the guidelines set by the Brazilian Council for Control of Animal Experimentation (CONCEA). Male C57BL/6 mice were bred and housed at the animal facility in the Department of Pharmacology, ICB/USP, under standard conditions (12 h light/dark cycle, temperature: 22 ± 1 °C, humidity: 65–75%) with ad libitum access to food and water.

### 2.2. Preparation and Exposure to the Pollutant 1,2-NQ and Exploring Pharmacological Treatments

#### 1,2-NQ Preparation and Animal Exposure

The 1,2-NQ aerosol spray was generated using a conventional ultrasonic nebulizer (Mod Respiramax; NS, Sao Paulo, SP, Brazil) operating at a flow rate of 1 mL/min, that was attached to a polycarbonate cage (surface area of 600 cm^2^) operating at a flow rate of 1 mL/min. The aerosol output within the cage had been previously characterized, as outlined in a previous study from our group [49]. The nebulized 1,2-NQ mass within the chamber was determined using the formula MWnq = Mnq × [NQ] × Vsol, where Mnq represents the nebulized mass of 1,2-NQ, MWnq is the molecular weight of 1,2-NQ (158.15 g/mol), [NQ] is the concentration of the pollutant solution, and Vsol denotes the volume of the pollutant solution placed in the nebulizer tube. The resulting concentration of 1,2-NQ nebulized within the chamber was approximately 152 ng, accounting for a 2–3% loss.

C57Bl6 male mice aged 42–43 days old were used for both in vivo and in vitro protocols after exposure to the pollutant 1,2-NQ or vehicle during the neonate period (Figure 1). Groups of 4 or 5 neonatal mice underwent exposure to the pollutant 1,2-NQ or its vehicle for 15 min on three alternate days (6th, 8th, and 10th days of life) [49]. Following each nebulization session, pups were returned to their mother’s cages. At 42 days of age, the mice were subjected to an in vivo electrocardiogram (ECG) test, with the mean arterial blood pressure (MABP) quantified. At 43 days of age, the mice were euthanized, and the right atria were removed for use in functional studies (Figure 1). In an additional set of experiments, neonatal mice were simultaneously treated with 1,2-NQ along with the TRPV1 antagonist capsazepine (50 mg/kg; i.p.) or the TRPC5 antagonist, ML-204 (2 mg/kg; i.p.) on days 6, 8, and 10.

1,2-NQ obtained from Sigma-Aldrich (Catalog Number 346616; CAS Number 524-42-5; MW: 158.15, St. Louis, MO, USA)) with a purity of ≥96.5% was used in the experiments. For nebulization, a fresh 100 nM solution of 1,2-NQ was prepared prior to use by dissolving 1,2-NQ in a solution comprising PBS (99.008%), Tween-80 (0.001%), and DMSO (0.001%), followed by a 5-min incubation in an ultrasonic bath, as previously outlined [49]. For experiments involving isolated atria, a 10 μM stock solution of 1,2-NQ was prepared by dissolving it in a PBS (50%) and DMSO (50%) solution, followed by a 5-min incubation in an ultrasonic bath. Subsequently, dilutions ranging from 10 to 7 and 10 to 3 were freshly prepared immediately before the experimental procedures.

Isoflurane was obtained from Cristalia (Itapira, SP, Brazil). Carbachol, isoproterenol, noradrenaline, forskolin, metoprolol, salbutamol, 1,2-NQ, ouabain, capsazepine, and ML-204 were obtained from Sigma Aldrich (St. Louis, MO, USA).

### 2.3. Evaluation of Cardiovascular Parameters

#### 2.3.1. Blood Pressure Measurement

At 42 days of age, systolic blood pressure was determined in trained, conscious mice from both the vehicle and 1-2-NQ-exposed groups using the indirect tail-cuff method (Power Lab 4/S, ADInstruments, Dunedin, New Zealand). The mice were warmed at 40 °C for 5 min, and three stable consecutive blood pressure measurements were taken and averaged [55].

#### 2.3.2. Heart Rate Variability (HRV)

The analysis of HRV serves as a dependable tool for evaluating cardiac autonomic modulation, predominantly influenced by the baroreflex [56]. The time domains of HRV were scrutinized by examining the variances between successive pulse intervals (PI) derived from electrocardiographic (ECG) signals recorded in anesthetized mice under 2.5% isoflurane anesthesia for a duration of 5 min. For ECG monitoring, electrodes were positioned on the animal’s skin following the Einthoven plane and base–apex, utilizing standard bipolar limb leads (I, II, and III). Data acquisition was carried out through a bioamplifier NeuroLog System (Digitimer, Hertfordshire, UK) with an A/D interface CED 1401 (CED, Cambridge Electronic Design, Cambridge, UK), employing Spike 2 software (CED, Cambridge Electronic Design, Cambridge, UK). HRV calculations were performed using the LabChart 8—HRV module (ADInstruments, Dunedin, New Zealand) and quantified by R-R interval (RR), and the square root of the sum of the squares of the differences between successive PI values (SDRR), and the square root of the mean of the sum of the squares of differences between adjacent NN intervals (RMSSD). Frequency domain analysis was executed through the Fast Fourier Transform method, which transformed R-R intervals into the frequency domain. Power spectra components were identified with LF as 0.20 to 0.75 Hz and HF as >0.75 Hz, with autonomic balance quantified as the LF/HF ratio [57].

### 2.4. Reactivity Assessment of Isolated Right Atria at 43 Days of Age

At the age of 43 days, mice previously subjected to 1,2-NQ or vehicle as neonates were anesthetized with 3% isoflurane, and an incision in the abdomen was made to collect blood from the abdominal aorta. The chest was then carefully opened, and the heart, along with the lungs, was excised and placed into a Petri dish containing Krebs’ solution containing (in mM) 115 NaCl, 25 NaHCO_3_, 4.7 KCl, 1.2 KH_2_PO_4_, 1.2 MgSO_4_-7H_2_O, 2.5 CaCl_2_-2H_2_O, 0.01 EDTA, and 11.1 dextrose, continuously aerated with 95% O_2_ plus 5% CO_2_ (*v*/*v*). For full details of the method setup, see the supplement. The heart and lungs were then transferred to a black silicon plate and attached with pins to facilitate the excision of the right atria. A tungsten wire (4–5 cm in length) was carefully inserted in the apical and distal regions of the atria. Subsequently, the atria were transferred to a wire-myograph chamber (Danish Myo Technology - DMT, Hinnerup, Denmark), attached to a transducer, and the temperature was set to 37 °C ± 0.2. A tension of 2.50 mN was applied to amplify the signal. Once the tension was set and the temperature stabilized, the atria were rinsed at 15-min intervals for 1 h until stabilized. The tissue was then utilized for experimental procedures.

### 2.5. Evaluation of Ouabain-Induced Fibrillation in Isolated Right Atria from Naive Mice

To assess the role of 1,2-NQ in the onset of arrhythmia, we employed a standardized in vitro fibrillation model [58,59]. Following stabilization, isolated right atria were exposed to 1,2-NQ (1 μM) and/or ouabain (10 μM). The onset of ouabain-dependent arrhythmias was continuously monitored over a 30-min period.

### 2.6. Statistical Analysis

All values are presented as mean ± standard error of the mean (SEM). The spontaneous beating of the right atria is expressed as beats per minute (BPM). Concentration–response curves were fitted using a sigmoid log equation with GraphPad Prism 8 (GraphPad Software Inc., San Diego, CA, USA), and the maximal BPM (Emax), as well as the negative logarithm of the concentration inducing a half-maximal response (pD2), were calculated. Intergroup differences in curves were compared using one-way ANOVA, and differences in other parameters between the groups were analyzed using a Student’s *t*-test. Statistical significance was set at *p* < 0.05.

## 3. Results

### 3.1. The Influence of Chemical Pollutant 1,2-NQ on Ouabain-Induced Arrhythmia in Isolated Right Atria

We first assessed the concentration–response curve of the air contaminant 1,2-NQ (10^−10^–10^−6^ M) on isolated right atria. The addition of 1,2-NQ resulted in a marked concentration-dependent positive chronotropic effect (Figure 2A). As expected, the administration of ouabain (10 μM) led to irregular contractions after 30 min. Notably, when 1,2-NQ (1 μM) was introduced along with ouabain, the onset of irregular contractions was reduced to 20 min. The positive chronotropic effect induced by 1,2-NQ alone corresponded to a significant (*p* < 0.05) increase in atrial contraction frequency compared to the control vehicle (Figure 2B). However, the addition of 1,2-NQ with ouabain did not affect ouabain-mediated changes in frequency (Figure 2C). These findings underscore the concentration-dependent ability of 1,2-NQ to induce positive chronotropism and its capacity to potentiate the arrhythmogenic effect of ouabain.

### 3.2. Neonatal Exposure to 1,2-NQ Impairs Heart Rate Variability but Does Not Affect Mean Arterial Blood Pressure

Figure 3 illustrates that early exposure to 1,2-NQ has no discernible impact on mean arterial blood pressure (MABP) compared to the control vehicle. However, the ECG data, as outlined in Table 1, reveals significant differences (*p* < 0.05) in HRV parameters, specifically RR, SDRR, and the LF/HF ratio, in mice exposed to 1,2-NQ as neonates compared to the control group (vehicle). These findings substantiate the notion that the air contaminant 1,2-NQ induces noteworthy alterations in heart function, potentially attributed to an augmentation in sympathetic nervous system activity. Although there were minor fluctuations in heart rate, these were not statistically significant (Table 1). These observations align with the hypothesis that 1,2-NQ may instigate arrhythmogenic events in the murine heart.

### 3.3. Neonatal Exposure to 1,2-NQ Impairs Adrenergic Responses in Isolated Right Atria

The functionality of the right atria was assessed in vitro, revealing that 1,2-NQ exerts a dose-dependent influence on the effects of noradrenaline (NE 10^−10^–10^−6^ M) on the rhythmicity of the right atrium. Mice subjected to 1,2-NQ during the neonatal stage exhibited a significant reduction in NE pD2 (6.72 ± 0.21 vs. 5.72 ± 0.27 * BPM; VEH and 1,2-NQ, respectively; *n* = 5/group, * *p* < 0.05) compared to the control vehicle group (Figure 4A). In contrast, the concentration–response to the muscarinic agonist carbachol (CCh 10^−10^–10^−6^ M) remained unaffected by neonatal treatment with 1,2-NQ (Figure 4B). These findings indicate that early exposure to 1,2-NQ may influence sympathetic tone.

### 3.4. The Contributions of β1-Adrenergic Receptor Signaling in Mediating 1,2-NQ-Induced Chronotropic Changes in the Right Atria

To elucidate the mechanisms underlying the impact of 1,2-NQ on the right atria, dose–response curves were constructed using isolated right atria and β-adrenergic agonists—noradrenaline (10^−10^–10^−6^ M; Figure 5A), isoproterenol (10^−10^–10^−6^ M; Figure 5B) and salbutamol (10^−10^–10^−6^ M; Figure 5C). Notably, 1,2-NQ (100 nM) hindered the effects of each agonist (Figure 5A–C; Table 2). Furthermore, the concentration–response curve to forskolin, known for increasing intracellular cAMP levels, failed to influence the 1,2-NQ-dependent positive chronotropism (Figure 6A, Table 2). In contrast, metoprolol (0.4 μM), a β1 antagonist, successfully impeded the positive chronotropic response induced by 1,2-NQ (Figure 6B, Table 2). These findings align with the notion that the presence of the contaminant 1,2-NQ diminishes the efficacy of β-adrenergic receptor signaling.

### 3.5. Inhibition of TRP Channels Disrupts the Positive Chronotropism Induced by Early-Life Exposure of 1,2-NQ

It has previously been suggested that TRP channels represent critical targets for the PM and its contaminant 1,2-NQ. In this study, we assessed the involvement of TRPV1 and TRPC5 in 1,2-NQ signaling. Prior to exposure to 1,2-NQ, mice underwent pretreatment with TRP antagonists. Capsazepine, a TRPV1 antagonist (CZP—50 mg/kg) [60], administered 30 min before 1,2-NQ exposure, successfully reversed the effects of this airborne contaminant on murine atria (Figure 7A, Table 3), without impacting the negative chronotropism induced by CCh (Figure 7B, Table 3). In a subsequent set of experiments, pretreatment with ML-204, a TRPC5 antagonist (2 mg/kg) [61,62], 30 min before 1,2-NQ exposure, significantly reduced the EMAX of the NE dose–response curve in mice, irrespective of exposure to 1,2-NQ (Figure 8A, Table 4). While changes were evident in positive chronotropism (Figure 8A), no discernible difference was observed in the negative chronotropism evoked by CCh (Figure 8B). These results highlight the significant contributions of TRP channels in the impairment of atrial function by exposure to 1,2-NQ.

## 4. Discussion

Air pollution poses a significant risk to heart health, specifically contributing to cardiovascular diseases through various compounds, with PM2.5 being a major component [63]. These particles originate from diverse sources such as soil, fossil fuel combustion, and industrial discharges. These tiny particles can linger in the air for long periods, making them easy for humans to inhale [64].

Even at levels below established air quality guidelines (e.g., <15 μg/m^3^—24-h mean), PM2.5 has been linked with high mortality rates [63,65]. Within the air, these PM particles absorb chemicals, like quinones [66,67], with concentrations of 13.7 μg of 1,2-NQ per g of DEP [66]. These substances are known to trigger specific immune responses, including airway inflammation [38,49,68] and oxidative stress through the production of reactive oxygen species (ROS) [47,48].

In this study, our research specifically examined the impact of premature exposure to a concentration of 1,2-NQ—10 μg/m^3^ inside a controlled chamber, below the WHO guidelines for PM2.5 (>15 μg/m^3^—24-h mean) on cardiac adrenergic function, specifically on the right atria contractility during the youthhood.

Our study in mice revealed that early exposure to nebulized 1,2-NQ poses a significant risk for developing cardiac adrenergic dysfunction in youth, potentially leading to atrial arrhythmias. Moreover, direct administration of 1,2-NQ in the isolated right atria of young mice elicited positive chronotropism, indicating that 1,2-NQ plays a role in the adverse cardiovascular events associated with PM2.5 exposure [69]. Consistent with these findings, studies on vulnerable aging populations have shown alterations in heart rate variability due to exposure to DEP, which contains chemical contaminants, including 1,2-NQ [70,71,72].

Our study revealed that the administration of metoprolol prevented 1,2-NQ-induced adrenergic dysfunction. This suggests that β1 receptor antagonists could be considered a potential mitigating or preventive therapy for individuals at cardiovascular risk related to PM or its contaminants, e.g., 1,2-NQ. Understanding the mechanisms by which 1,2-NQ induces atrial dysfunction in mice may pave the way for similar investigations in humans. Furthermore, numerous meta-analyses and epidemiological studies have consistently highlighted the association between increased levels of AP, including PM, and a higher risk of developing cardiovascular disturbances [7,69]. For potential signaling pathways, see Appendix A.

Our findings underscore the detrimental impact of air pollution components, particularly 1,2-NQ, on cardiac adrenergic function, emphasizing the potential for targeted therapies and the importance of further investigations into these mechanisms in both animal models and human populations.

Heart alterations (sympathetic tone unbalancing) in ECG and BP are commonly associated with PM exposure [71,72]. We observed that neonatal exposure to 1,2-NQ elevates the risk of inducing alterations in atrial rhythmicity, independently of promoting impairment in blood pressure in the young adult population. These findings were unexpected as the elevation in BP is commonly associated with an increase in sympathetic tone, affecting atrial rhythmicity.

The co-administration of the cardiac glycoside ouabain with 1,2-NQ increased the contraction force. Such findings reinforce the understanding that this PM-contaminant potentiates cardiovascular events such as arrhythmia in a direct way and reinforces the role played by 1,2-NQ as one of the players in inducing AP-mediated arrhythmogenic events.

According to pre-clinical studies, the mechanisms involved in the cardiovascular changes caused by PM are complex and vary according to the exposure protocol used. The inhalation of PM2.5 leads PM2.5 to penetrate the airways, where it is translocated via alveoli to the circulation [42,73]. In both the lung and cardiovascular systems, PM induces an inflammatory response due to the generation and release of pro-inflammatory mediators, as well as ROS/RNS production [42,73]. Other studies have demonstrated that PM can accumulate in blood vessels, increasing the risk of thrombus, heart disease, and stroke [73,74]

The ambient pollutant PM2.5 has been associated with an increase in atrial fibrillation onset within hours following exposure [71]. However, the mechanisms associated with proarrhythmic events induced by PM contaminants, including 1,2-NQ, are unclear despite several reviews on this topic [75,76]. We, therefore, investigated the mechanisms associated with the impaired atrial rhythmicity induced by precocious exposure to 1,2-NQ. Based on the significant reduction of noradrenaline (NE)-mediated BPM response as well as reduced NE pD2 values, we revealed a detrimental role of premature life exposure to 1,2-NQ in the programming of youthhood atrial dysfunction in which the adrenergic receptor seems to be a potential target for 1,2-NQ in the right atria. Adversely affecting atrial rhythmicity (BPM) upon early life exposure to 1,2-NQ is in agreement with our in vitro data of 1,2-NQ exposure in which the incubation of mouse isolated right atria with 100 nM of this pollutant markedly reduced both BPM response and pD2 values to increasing concentrations of NE and ISO, and to a lesser extent salbutamol (β2-adrenergic agonist) pD2 value, thus reinforcing the suggestion that the β1-adrenoceptor pathway is strongly affected by 1,2-NQ. This is further confirmed by the findings that the positive chronotropism elicited by increasing concentration of 1,2-NQ in the isolated right atria was abolished in the presence of metoprolol, a selective β1-adrenoceptor antagonist. Of interest, in a previous study using a similar exposure approach with concentrated PM2.5, the authors showed that mice exposed to PM2.5 in the uterus showed low birth weight and, in adulthood, exhibited both reduced left ventricular fractional shortening and pressure–volume loops ejection fraction in addition to increased end-systolic volume and reduced dP/dt maximum and minimum when compared to controls [69]. Au and co-workers [77] showed that the resting atrial rate doubled in response to adrenaline sensitivity between 1 and 20 days of a rat’s life, thus suggesting a maturational correlation in atrial β-adrenoceptors. In addition, they showed that in 2- and 20-day-old rat atria, the force increases with different adrenaline concentrations correlating linearly with the log of the increase in cyclic AMP. Furthermore, the in vitro exposure of bovine aortic endothelial cells to 1,2-NQ evoked covalent modification of cAMP response element-binding protein (CREB), thereby inhibiting its DNA binding activity and substantial gene expression of B-cell lymphoma-2 (Bcl-2) regulated by this transcription factor [78]. Later, the same authors showed that the mediated-covalent interaction between CREB and 1,2-NQ through Cys-286 blocks the DNA binding activity of CREB, resulting in the repression of CREB-regulated genes [79]. Knowing that a variety of environmental pollutants, including 1,2-NQ (or its metabolites), may cause abnormal DNA methylation, which further disturbs gene expression [80,81,82,83], it is possible that due to the chemical nature of 1,2-NQ (and its metabolite) that displays cytotoxic and genotoxic properties. This could include increasing topoisomerase II-mediated double-stranded DNA scission [84,85] and non-enzymatic oxidative transformation of 1,2-NQ to form an intermediate PAH-epoxide, which covalently binds to DNA [80]. However, more research is required to elucidate the 1,2-NQ-mediated mechanisms that affect right atria maturation or improper atria or heart development.

Over the last few years, evidence has linked transient receptor potential channels, mainly TRPV1 and TRPA1 channels, as potential targets for AP-induced risk factors/side effects, particularly in the airways [86,87,88] and to a lesser extent in the cardiovascular system [28,89,90]. A number of studies provided robust evidence that different-sized PM from traffic-related air pollution increases intracellular calcium levels by interacting with calcium channels like TRP channels, G protein-coupled receptors such as β-adrenoceptors, TLR4, and protease-activated receptor 2 [PAR-2]) [90,91,92]. However, little is known about TRP channel involvement in 1,2-NQ-induced atrial disturbances.

We proceeded to investigate the interaction of early acute exposure to 1,2-NQ with TRPV1 channels, mainly based on our previous evidence that acute inhalation of 1,2-NQ in addition to DEP promoted increased gene expression of TRPV1 in rat bronchus [38]. When applied in isolated guinea trachea, 1,2-NQ activated TRPV1 through increased protein tyrosine phosphorylation, leading to the contraction of this tissue. Interestingly, TRPV1 expression is linked to cardiac hypertrophy, while its knockdown protected heart function in a murine model of hypertrophy [93]. Chronic cardiac TRPV1 afferent signaling following myocardial infarction promotes arrhythmogenic events [94]. Kumagai and colleagues identified TRPV1 as a significant target for 1,2-NQ in the upper airways [95]. In agreement, we showed that the pharmacological blockade of TRPV1 prior exposure to 1,2-NQ or its vehicle led to a significant increase in NE pD2, reversing the actions of 1,2-NQ as evaluated in the isolated right atria. This observation supports previous findings demonstrating that TRPV1 is a target for 1,2-NQ and confirms that exposure to 1,2-NQ can be one of the triggers for arrhythmogenic events within PM.

A potential additional target that contributes to the youthhood consequences (right atria impairment) of premature exposure to 1,2-NQ is the canonical TRPC5, as these channels have been involved in cardiac plasticity [96,97], as well as being a target for heavy metals [98]. This has not been the case in the current study, as the pharmacological blockade of TRPC5 did not reverse 1,2-NQ-induced atrial dysfunction. Instead, it has increased sensitivity to cholinergic stimuli and greatly reduced BPM in response to the NE in the right atrium of mice previously exposed to 1,2-NQ. Of note, the pharmacological treatment of vehicle-exposed mice with the TPRC5 antagonist per se significantly reduced the basal adrenergic function (reduced BPM), thus showing a clear linking of TRPC5 with the right atria autonomic physiological dynamic. In the presence of 1,2-NQ exposure, the TRPC5 antagonist ML204 further reduced the atria rate (BPM), but it did not markedly affect carbachol-mediated changes in the percentage of BPM.

Although Gq-coupled M1-receptors seem to regulate human atrial IK, Ach, which is likely to be augmented in patients with long-standing persistent atrial fibrillation [99], we did not reveal a clear insight into the implication of the M1-receptor in the 1,2-NQ-mediated right atria disturbances.

## 5. Conclusions

In conclusion, our findings show for the first time that (neonatal) mice exposed to the chemical pollutant 1,2-NQ face an elevated risk of developing cardiac adrenergic dysfunction as young adults. This physiological response seems to be linked to the potential activation of TRPV1 channels, consequently fostering an augmentation in β1-adrenergic receptor-mediated atrial rhythm. These results allow an understanding of potential mechanisms via which 1,2-NQ exposure may induce detrimental effects on heart function (Figure 9).

## Figures and Tables

**Figure 1 biomolecules-14-00057-f001:**
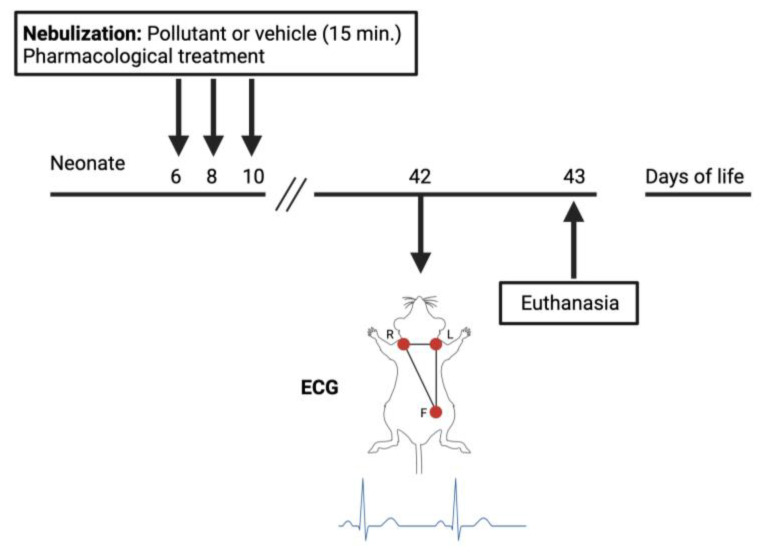
Schematic overview of experimental design for premature exposure to pollutant 1,2-NQ and cardiovascular assessments.

**Figure 2 biomolecules-14-00057-f002:**
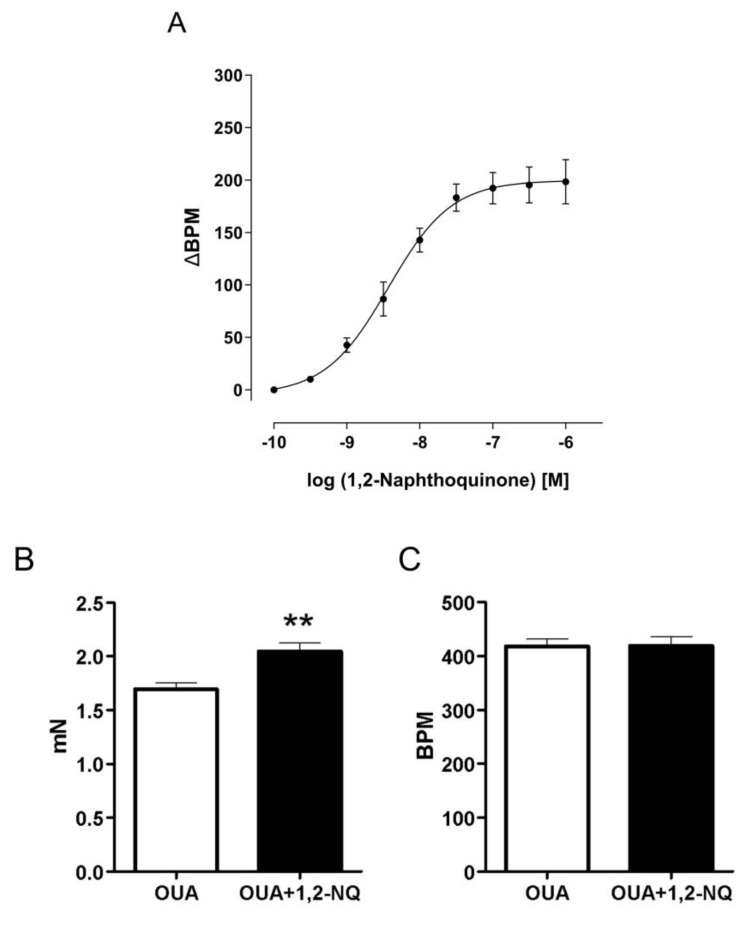
Impact of 1,2–NQ on isolated murine right atria. (**A**) The concentration–response curve of 1,2–NQ induces positive chronotropism in isolated murine right atria. (**B**) The atrial force contraction triggered by ouabain (OUA, white column) demonstrates a significant increase in the presence of 1,2–NQ (100 nM; black column). (**C**) 1,2–NQ does not alter ouabain-induced atrial frequency (BPM). ** *p* < 0.05 vs. ouabain. BPM (beats per minute), 1,2–NQ - 1,2–naphthoquinone.

**Figure 3 biomolecules-14-00057-f003:**
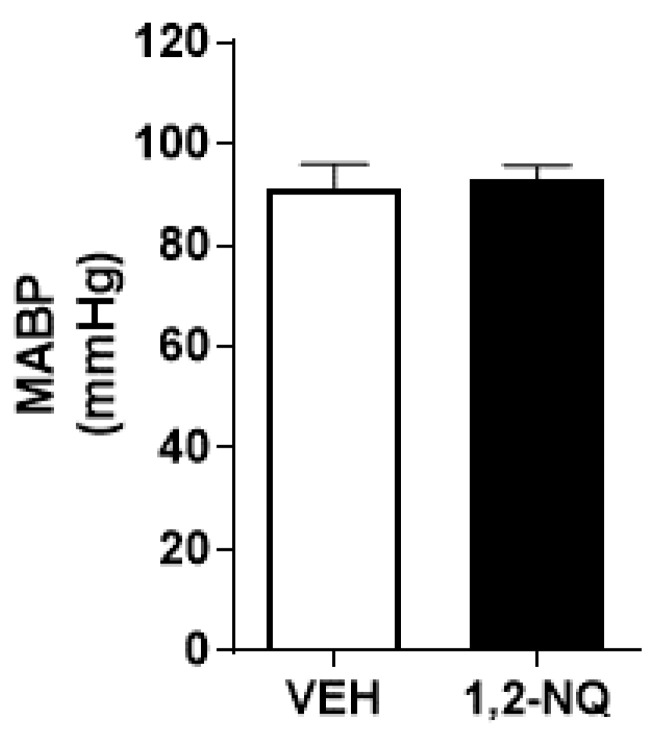
Impact of 1,2-NQ on mean arterial blood pressure. Exposure to 1,2-NQ during the neonatal stage (6, 8, and 10th of life) does not affect mean arterial blood pressure compared to the vehicle group (white column). Data are expressed as mean ± s.e.m. for *n* = 5 in each group.

**Figure 4 biomolecules-14-00057-f004:**
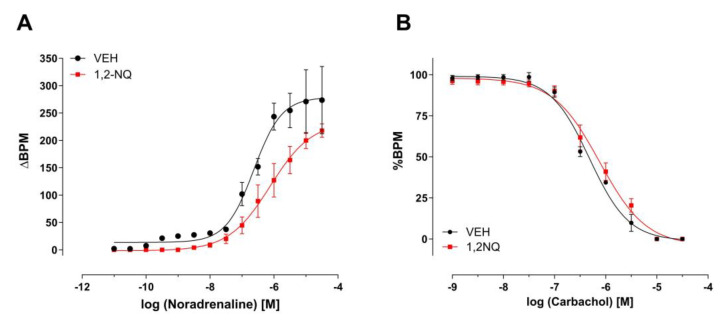
Impaired atrial rhythmicity in young mice previously exposed to 1,2–NQ as neonates. In panel (**A**), neonate exposure to 1,2–NQ (red squares) results in impaired atrial rhythmicity evoked by noradrenaline compared to the control vehicle group (black circles). In panel (**B**), the negative chronotropic response of carbachol remains unaffected by premature exposure of 1,2–NQ (red squares) compared to the control vehicle (black circles). Data expressed as mean ± s.e.m. for *n* = 5 in each group. Heart rate changes (ΔBPM) are presented as beats per minute evoked by noradrenaline, while %BPM represents beats per minute evoked by carbachol. 1,2–NQ - 1,2–naphthoquinone.

**Figure 5 biomolecules-14-00057-f005:**
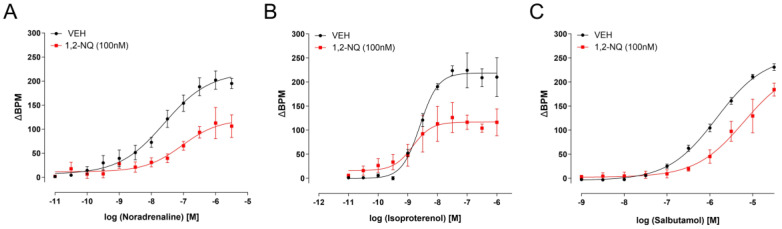
Impaired atrial rhythmicity with 1,2–NQ incubation in isolated right atria. In comparison to the vehicle (black circles), incubation with 1,2–NQ (100 nM; red squares) adversely affects atrial rhythmicity in response to noradrenaline (**A**), isoproterenol (**B**), and salbutamol (**C**) concentration–response curves. The data are expressed as mean ± s.e.m. for *n* = 5 in each group. BPM (beats per minute) is expressed as the rate evoked by the agonist. 1,2–NQ — 1,2–naphthoquinone.

**Figure 6 biomolecules-14-00057-f006:**
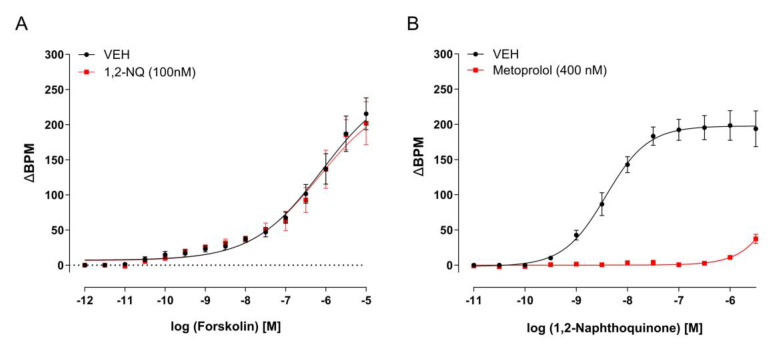
Impact of 1,2–NQ on forskolin-induced positive chronotropism and effects of metoprolol on 1,2-NQ-induced increased BPM in isolated right atria. Panels (**A**) illustrate that the positive chronotropic effect induced by forskolin remains unaffected by 1,2–NQ (100 nM; red squares) compared to the vehicle (black circles). Conversely, panel (**B**) shows that metoprolol incubation (400 nM; red squares) completely inhibits the positive chronotropic response evoked by 1,2-NQ when compared to vehicle control (black circles). Data are expressed as mean ± s.e.m. for *n* = 5 in each group. BPM (beats per minute). 1,2–NQ—1,2–naphthoquinone.

**Figure 7 biomolecules-14-00057-f007:**
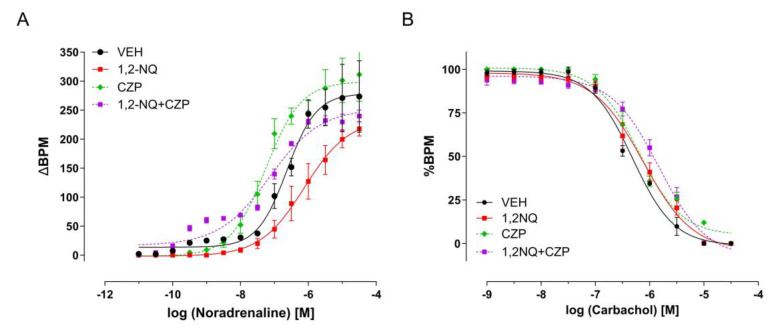
Reversal of atrial rhythmicity impairment in young mice previously exposed to 1,2–NQ by capsazepine, a TRPV1 antagonist. Concentration–response curves for noradrenaline (**A**) and carbachol (**B**) were conducted on isolated right atria from the following groups: Vehicle (black circles), 1,2–NQ (red squares), capsazepine (CZP, green diamonds), and 1,2–NQ + CZP (purple squares). The data is expressed as mean ± s.e.m. Heart rate (BPM) is presented as beats per minute evoked by noradrenaline, and %BPM is presented as beats per minute evoked by carbachol. 1,2–NQ – 1,2–naphthoquinone.

**Figure 8 biomolecules-14-00057-f008:**
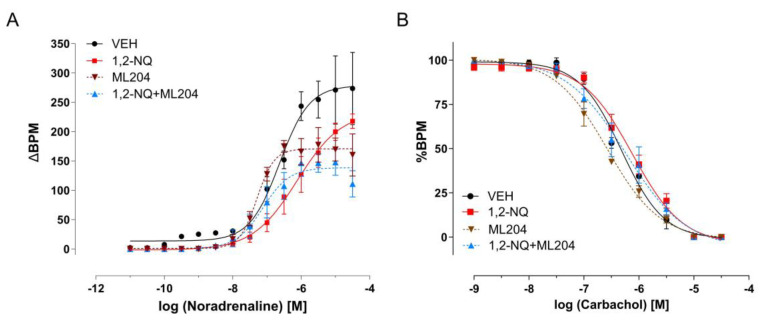
Blockade of TRPC5 on 1,2–NQ-induced atrial rhythmicity impairment. Concentration–response curves to noradrenaline (**A**) and carbachol (**B**) were performed on isolated right atria from the following groups treated as a neonate with vehicle (black circles), 1,2–NQ (red squares), TRPC5 antagonist (ML204, brown triangles), and 1,2–NQ plus ML204 (blue triangles). Data are expressed as mean ± s.e.m. for *n* = 5 in each group. Beats per minute (BPM) are presented as responses to noradrenaline, while %BPM represents responses to carbachol. 1,2–NQ—1,2–naphthoquinone.

**Figure 9 biomolecules-14-00057-f009:**
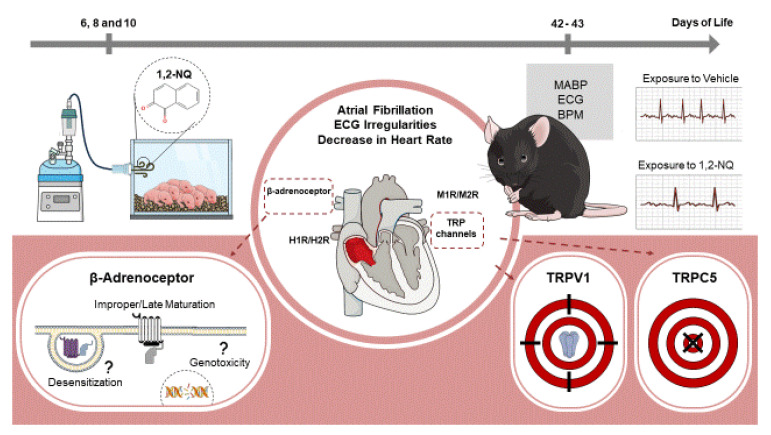
Early-life exposure to 1,2-naphthoquinone (1,2-NQ) significantly heightens the likelihood of cardiac adrenergic dysfunction, manifesting as atrial fibrillation, electrocardiogram irregularities (ECG), and diminished heart rate. This heightened risk potentially predisposes individuals to atrial arrhythmia during their youth. The underlying mechanisms hinge on the potential disruption of β-adrenoceptors and the impairment of atrial maturation or proper heart development (desensitization or genotoxicity). This process is partially mediated by TRPV1 activation but not by TRPC5.

**Table 1 biomolecules-14-00057-t001:** Analysis of heart rate variability in mice exposed to 1,2-naphthoquinone or its vehicle. Values are expressed as mean ± sem. * *p* < 0.05 vs. vehicle.

	Vehicle (*n* = 6)	1,2-NQ (*n* = 7)
***RR*** **(ms)**	100.0 ± 3.4	91.3 ± 1.3 *
***SDRR*** **(ms)**	4.1 ± 0.5	2.4 ± 0.2 *
***RMSSD*** **(ms)**	1.0 ± 0.2	0.7 ± 0.1
** *LF/HF ratio* **	0.13 ± 0.01	0.34 ± 0.07 *
***Heart rate*** **(BPM)**	612 ± 18	650 ± 12

**Table 2 biomolecules-14-00057-t002:** Maximal response (EMAX) and pD2 calculated from concentration–response curves for noradrenaline (NE), salbutamol (SB), and isoproterenol (ISO) in isolated right atria from mice exposed to 1,2-naphthoquinone or its vehicle. Values are expressed as mean ± sem. * *p* < 0.05 vs. vehicle. *n* = 5 in each group.

	EMAX (BPM)		pD2	
	VEH	1,2-NQ	VEH	1,2-NQ
**NE**	197 ± 11	112 ± 12 *	7.73 ± 0.15	7.12 ± 0.25 *
**SB**	254 ± 8	233 ± 30	5.84 ± 0.05	5.20 ± 0.18 *
**ISO**	200 ± 7	111 ± 7 *	8.66 ± 0.09	8.87 ± 0.22 *
**FK**	261 ± 29	245 ± 20	6.11 ± 0.25	6.18 ± 0.18

**Table 3 biomolecules-14-00057-t003:** Maximal response (EMAX) and pD2 were calculated from concentration–response curves to NE and CCh in isolated right atria from mice exposed to 1,2-NQ and treated with capsazepine. Values are expressed as mean ± sem. * *p* < 0.05 vs. Vehicle, ^#^ *p* < 0.05 vs. 1,2-NQ, ^$^ *p* < 0.05 vs. CZP. *n* = 5 in each group.

	VEH	1,2-NQ	CZP	1,2-NQ + CZP
**NE**				
**EMAX (BPM)**	259 ± 27	218 ± 12	313 ± 43	242 ± 9
**pD_2_**	6.63 ± 0.10	5.73 ± 0.2 *	7.26 ± 0.10 *^#^	7.14 ± 0.10 *^#^
**CCh**				
**pD_2_**	6.29 ± 0.04	6.12 ± 0.06	6.20 ± 0.05	5.83 ± 0.05 *^$^

**Table 4 biomolecules-14-00057-t004:** Maximal response (EMAX) and pD2 were calculated from concentration–response curves to NE and CCh in isolated right atria from mice exposed to 1,2-NQ and treated with ML-204. Values are expressed as mean ± sem. * *p* < 0.05 vs. Vehicle, ^#^ *p* < 0.05 vs. 1,2-NQ, ^$^ *p* < 0.05 vs. ML-204. *n* = 5 in each group.

	VEH	1,2-NQ	ML-204	1,2-NQ + ML-204
**NE**				
**EMAX (BPM)**	259 ± 27	218 ± 12	170 ± 10 *^#^	138 ± 9 *^#^
**pD_2_**	6.63 ± 0.10	5.73 ± 0.20 *	7.29 ± 0.10 *^#^	7.12 ± 0.10 *^#^
**CCh**				
**pD_2_**	6.29 ± 0.04	6.12 ± 0.06	6.57 ± 0.04 *^#^	6.23 ± 0.05 ^$^

## Data Availability

The data presented in this study are available on request from the corresponding authors.

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
