# Peer review of "Disruption of Atrial Rhythmicity by the Air Pollutant 1,2-Naphthoquinone: Role of Beta-Adrenergic and Sensory Receptors"

_biomolecules, 2023, doi:10.3390/biom14010057_

Round 1
Reviewer 1 Report
Comments and Suggestions for Authors
1. The section discussing the mechanistic basis of PM2.5-induced cardiovascular disturbances is detailed but complex. Consider simplifying the language where possible without sacrificing accuracy to improve accessibility for a broader audience.
2. Review the document for grammatical errors and punctuation issues. For instance, in some sentences, there are missing commas or awkward phrasing that could be improved for better clarity.
3. Consider incorporating relevant figures or tables to visually represent key data or relationships. This could enhance the understanding of complex information, especially in the mechanistic basis section.
4. Provide more details about the composition of the exposure chamber during the inhalation of 1,2-NQ. Specify the type of nebulizer used, the volume of the exposure chamber, and any additional parameters relevant to the inhalation process.
5. Provide additional information on the specific chemicals used, especially if there are variations or specific preparations. For example, mention the purity of the chemicals and any specific considerations for their use in the study.
6. Clearly outline the steps involved in the experimental procedures, especially for the reactivity assessment of isolated right atria and the evaluation of ouabain-induced fibrillation. Provide step-by-step details for each stage of the experiment.
7. Include a brief justification or rationale for choosing specific methods and concentrations, especially in relation to the exposure concentration of 1,2-NQ. Explain why the chosen concentration is relevant to the study's objectives.
8. Explore the cellular and molecular mechanisms underlying the observed changes. For example, conducting experiments to examine intracellular signaling pathways affected by 1,2-NQ could deepen the understanding of how this pollutant influences cardiac function.
9. Extend the concentration-response analysis to explore a broader range of concentrations for 1,2-NQ. This could help establish a more comprehensive dose-response relationship and identify potential threshold levels for adverse effects.
10. Connect the findings more explicitly to clinical relevance. Discuss how the observed effects in mice may translate to human cardiovascular health and whether the concentrations used in the study are within a range relevant to human exposure.
11. The results involving TRP channel antagonists (TRPV1 and TRPC5) are intriguing. Expanding on the mechanistic understanding of how these specific channels are involved in the impairment of atrial function could be beneficial. Consider planning experiments that delve deeper into the intracellular signaling pathways influenced by TRPV1 and TRPC5 activation. This could involve examining downstream effectors and potential cross-talk with other signaling cascades.
12. Investigating the temporal dynamics of 1,2-NQ effects is crucial. Perform experiments at different time points post-exposure to assess whether the observed alterations in atrial function are transient or persistent. This could provide insights into the recovery or adaptation mechanisms of the cardiovascular system following early-life exposure to 1,2-NQ. Consider examining whether there are critical periods during development when the heart is more susceptible to the effects of this pollutant.
Comments on the Quality of English LanguageModerate editing of English language required
Author Response
Dear Editor,
Thanks a lot for your comments concerning our work, which were invaluable for improving the quality of the manuscript. We conformed to most of the inquires raised by this reviewer. Most of the answers are detailed in the attached document, and more concise description was included in the manuscript due to the word number limitation.
"Please see the attachment".

Reviewer 2 Report
Comments and Suggestions for Authors
Authors present an interesting study of the effect of 1,2-NQ on cardiac function and atrial tissue. However, there are three major issues and a few minor one.
Major
1. Introduction is two broad and unfocused. I think should be more focused on PM2.5 (what is it? it was not explained until Discussion) and cardiovascular effects.
2. I am not sure that claim of prevention of the effect of 1,2-NQ by CZP can be made on the basis of the existing results because 1,2-NQ+CZP response is bi-phasic. At least, the claim need to be more carefully worded.
3. What is the direct effect of 1,2-NQ on the atrial force generation? It may clarify TRPV1 involvement one way or the other.
Minor
(see the markup in the attached pdf)

Quality of English language is satisfactory. I believe I made only handful of language related corrections
Author Response
Dear Editor,
Thanks a lot for your comments concerning our work, which were invaluable for improving the quality of the manuscript. We conformed to all of the inquires raised by this reviewer, and most of the answers are detailed in the attached document.
"Please see the attachment".

Round 2
Reviewer 1 Report
Comments and Suggestions for Authors
No comments
Comments on the Quality of English LanguageMinor editing of English language required